# LeanProgress: Guiding Search for Neural Theorem Proving via Proof Progress Prediction

**Robert Joseph George**                                         *rgeorge@caltech.edu*
*Computing + Mathematical Sciences Department*
*California Institute of Technology*

**Suozhi Huang**                                         *suozhi.huang@princeton.edu*
*Computer Science Department*
*Princeton University*

**Peiyang Song**                                         *psong@caltech.edu*
*Computing + Mathematical Sciences Department*
*California Institute of Technology*

**Animashree Anandkumar**                                         *anima@caltech.edu*
*Computing + Mathematical Sciences Department*
*California Institute of Technology*

**Reviewed on OpenReview:** *https://openreview.net/forum?id=eTmOwvvRu9*

## Abstract

Mathematical reasoning remains a significant challenge for Large Language Models (LLMs) due to hallucinations. When combined with formal proof assistants like Lean, these hallucinations can be eliminated through rigorous verification, making theorem proving reliable. However, even with formal verification, LLMs still struggle with long proofs and complex mathematical formalizations. While Lean with LLMs offers valuable assistance with retrieving lemmas, generating tactics, or even complete proofs, it lacks a crucial capability: providing a sense of proof progress. This limitation particularly impacts the overall development efficiency in large formalization projects. We introduce LeanProgress, a method that predicts the progress in the proof. Training and evaluating our models made on a large corpus of Lean proofs from Lean Workbook Plus and Mathlib4 and how many steps remain to complete it, we employ data preprocessing and balancing techniques to handle the skewed distribution of proof lengths. Our experiments show that LeanProgress achieves an overall prediction accuracy of 75.8% in predicting the amount of progress and, hence, the remaining number of steps. When integrated into a best-first search framework using Reprover, our method shows a 3.8% improvement on Mathlib4 compared to baseline performances of 41.4%, particularly for longer proofs. These results demonstrate how proof progress prediction can enhance both automated and interactive theorem proving, enabling users to make more informed decisions about proof strategies. Our code is merged in this library here `https://github.com/lean-dojo/LeanDojo-v2`.

## 1 Introduction

Formal theorem proving (Avigad, 2023) has emerged as a cornerstone of rigorous mathematical verification, providing machine-checked guarantees for proofs ranging from foundational results (Gowers et al., 2023) to industrial applications (Community, 2022). The Lean proof assistant (Moura & Ullrich, 2021), built on dependent type theory, has witnessed remarkable adoption growth (Best et al., 2023), fueled by collaborative efforts on large-scale formalization projects (mathlib Community, 2020) and novel mathematical develop-

ments (Asgeirsson, 2024). This collaborative paradigm shift underscores the urgent need for enhanced tooling to support mathematicians navigating increasingly complex proof environments.

The recent success of Large Language Models (LLMs) in code generation (Rozière et al., 2024) and symbolic reasoning (Yu et al., 2024) has spurred innovations at the intersection of LLMs and formal verification. Some of the works that have been developed include LeanDojo (Yang et al., 2024b) which provides an interactive environment for training LLMs on tactic-level interactions while LLMStep (Welleck & Saha, 2023) and LeanCopilot (Song et al., 2024) focuses on next-tactic suggestion through interface as a useful tool. Lean Agent (Kumarappan et al., 2024) then combines neural suggestion with life-long learning while Lean Aide (Agrawal et al., 2022) and Lean-STaR(Lin et al., 2024) translates statements written in natural language in a doc-string like format to Lean types (including theorem statements) and bootstrapping thoughts. While these systems demonstrate impressive tactic-level accuracy (Johansson, 2023), they primarily optimize for local correctness rather than global proof progress – a critical limitation when navigating Lean's vast action space (Nawrocki et al., 2023).

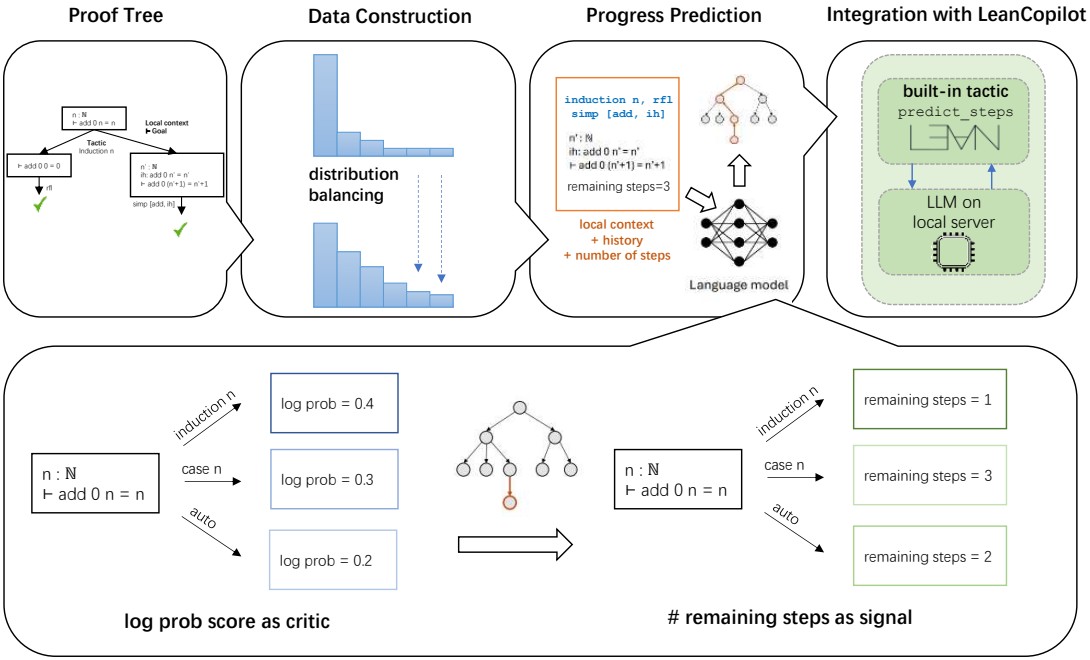

Figure 1: **The visualization of LeanProgress.** LeanProgress is a lightweight framework that collects the number of remaining steps in proof trees and then balances the data distribution to train the language model. Then LeanProgress takes the proof state as input to generate the remaining steps for each state as a signal for search. LeanProgress also integrates the tactic `predict_steps` in LeanCopilot as a user-friendly tool.

Reinforcement learning (RL) presents a theoretically appealing framework for automated theorem proving (Dong et al., 2024), where finding reward signals over proof trajectories is essential. However, the combinatorial explosion of tactic sequences in Lean (Clune, 2023) renders direct RL applications impractical (Setlur et al., 2024). Alphaproof (Hubert et al., 2025) has done RL for theorem proving but it's not open source and needs enormous compute. Current approaches mitigate this through hybrid architectures (Wang et al., 2023) but remain fundamentally limited by the absence of reliable progress indicators to guide exploration – a prerequisite for effective RL in mathematical domains (Gao et al., 2024).

We address this critical gap with **LeanProgress** (Fig 1), a lightweight framework that predicts remaining proof steps through learned progress signals with search methods beyond log-probability based search (Song et al., 2024) or manual heuristics (Ringer et al., 2021).

LeanProgress makes the following key contributions: **Balanced Data Generation Pipeline:** We construct a balanced dataset of approximately 80k proof trajectories from Lean Workbook Plus and Mathlib4 by performing a tree search and selecting the shortest path as ground truth. We employ a data balancing strategy based on relative proof progress. Since the useful and non-trivial data of long proofs are long-tailed distributed in the original dataset, we fully utilize long proof data by assigning each state with a remaining step as a label.

**Model for Progress Prediction:** We fine-tune a DeepSeek Coder V1 1.3b base model to predict the remaining steps, achieving a Mean Absolute Error (MAE) of 3.15 and an overall prediction accuracy of 75.8% on the test set with proof history. Unlike tactic suggestion tools, LeanProgress provides a global view of the proof process by predicting the remaining steps rather than the immediate next tactic.

**Progress-Guided Proof Search:** We integrate our step prediction model into a best-first search framework. A natural first step for using the progress predictor is combining the predicted remaining steps with the tactic generator's log probabilities to guide the search. In the future, we hope to use this, instead of just relying on the log probabilities, as a reward for RL. We observe on Mathlib4 a significant improvement of 3.8% with the baseline Reprover performance of 41.4%.

**Integration with LeanCopilot:** Based on the LeanCopilot framework, we provide a new built-in tactic `predict_steps_with_suggestion` within the standard Lean user interface. It is a helpful tool that not only suggest tactics but also offer users immediate feedback on proof progress and potential next steps. All the development details are in Appendix B and C.

## 2 Related Work

**LLMs for Formal Proof Generation.** Large Language Models (LLMs) have demonstrated significant potential in the field of formal theorem proving (Yang et al., 2024a), finding applications across various proof assistants (Yang et al., 2024b; Song et al., 2024; Lama et al., 2024). Current research on LLM-based theorem proving primarily focuses on several key tasks. A prominent application is tactic suggestion. Following GPT-f (Polu & Sutskever, 2020), LLMs are employed to predict the most promising next tactic given the current proof state. These methods are often coupled with proof search algorithms, such as best-first search (Yang et al., 2024b) or majority voting (Zhou et al., 2024), to explore the proof space and discover complete proofs (Wu et al., 2024). Other techniques, such as retrieval-augmented LLMs (Yang et al., 2024b) and agentic approaches (Thakur et al., 2024), provides further aids for tactic generation by selecting relevant lemmas and enabling multi-round proof refinement utilizing environment feedback. Moreover, emerging research directions include autoformalization (Wu et al., 2022; Jiang et al., 2023), which aims to translate informal mathematical text into formal proofs, and the direct generation of complete proof sketches (Jiang et al., 2022; Wang et al., 2024), both of which can be combined with proof generation to enable large-scale training despite inherent proof data scarcity. Our work addresses the gap from local tactic prediction to a global understanding of the proof trajectory by focusing on predicting the number of remaining steps required for proof completion, offering a novel way for new applications of reinforcement learning in automated theorem proving.

**Interactive Tools for Formal Theorem Proving.** Mathematicians proving theorems in Lean can significantly benefit from interactive tools that integrate seamlessly into the Lean workflow and provide aids. LLMStep (Welleck & Saha, 2023) extracts current proof states from Lean and sends it to a remote server for LLM-generated tactic suggestions. LeanCopilot (Song et al., 2024) improves the user experience by having fully native tactic suggestion and proof search tools in Lean, besides an additional functionality of premise selection, providing more comprehensive assistance for the proving process. CoqPilot (Kozyrev et al., 2024), a VS Code extension for Coq, uses LLMs, among other generative methods, to fill in proof holes by an "admit" tactic. Unlike these tactics or proof-centric approaches, we predict the number of remaining steps by

adding a new tactic, `predict_steps_with_suggestion` based on LeanCopilot, providing tactic suggestions ranked by the output number of remaining steps as a score.

**LLM Guidance in Search.** Effective proof search is essential for automated theorem proving. While scaling computational resources during search has led to significant advancements, as seen in AlphaGeometry (Trinh et al., 2024) and AlphaProof (Hubert et al., 2025) for IMO problems and in recent work on natural language reasoning (Lightman et al., 2023; Yang et al., 2022; Zhang et al., 2024; Xie et al., 2024) (including OpenAI's o1, o3 model (Jaech et al., 2024; Xu et al., 2025)), proof search is a bit different. The vast search space of possible proof steps necessitates effective guiding mechanisms. This highlights the need for methods that can provide a global perspective on proof progress, which our work addresses by predicting the number of remaining steps.

**Progress Signals for Proof Search.** Several systems learn value-like guidance signals for proof search. GamePad (Huang et al., 2019) introduces a learning environment for Coq and studies both tactic prediction and position evaluation, producing estimates of how promising a proof state is. More recently, QEDCartographer (Sanchez-Stern et al., 2025) uses reinforcement learning to learn a proof-state evaluation signal tailored to the branching structure of proofs, improving automated proof synthesis in Coq. LeanProgress targets a complementary notion of guidance: a supervised, interpretable distance-to-goal signal by predicting the remaining number of steps to completion from successful traces. Unlike generic value estimates, this output is directly interpretable as progress, and it composes naturally with existing tactic generators (e.g., as a lightweight reranking prior alongside log-likelihood).

# 3 Data Generation for LeanProgress

This section details the data generation and processing methodology used to train LeanProgress. We describe the process of generating proof trees using best-first search and the Reprover model, the resulting dataset of proof trajectories, and the adjustments made to address the skewed distribution of proof lengths.

## 3.1 Preliminaries: Tactic Prediction as an MDP

Interactive Theorem Provers (ITPs) frame theorem proving as a search problem. As Fig 2 shows, the initial theorem to be proven represents the initial state, and the application of tactics generates transitions to new states, each containing subgoals. The objective is to find a sequence of tactics that leads to a state where all subgoals are proven. This search process is central to automated theorem proving, and our work focuses on providing valuable information to guide this search within the Lean ITP.

The theorem-proving problem can be formalized as a Markov Decision Process (MDP), denoted as $(S, A, P_a, R_a)$, where $S$ represents the set of all possible proof states. $A$ represents the set of all available tactics (actions). $P_a$ represents the state transition probabilities after executing tactic $a$ in state $s$. $R_a$ represents the reward obtained by executing tactic $a$. From an MDP perspective, a proof process can be viewed as a trajectory of states, tactics, and rewards: $(s_i, a_i, r_i)$, where the proof assistant (e.g., Lean) provides the next state $s_{i+1}$ given the current state $s_i$ and the applied tactic $a_i$. In typical tactic prediction, proving a theorem involves providing a proof state $s$ to a language model $L$, which then generates a tactic $a$, i.e., $\pi_L(a|s)$. Typically, final states (where the goal is proven) are assigned a reward of 1, indicating successful completion of the proof.

## 3.2 Generating Proof Trees and Trajectories

A common evaluation strategy for neural theorem provers is best-first search, as used in GPT-f and related research (Han et al., 2021). This method explores the proof space by iteratively expanding the "best" state, determined by the maximum cumulative log probability of the preceding proof trajectory. Specifically, given a set of unexpanded states $s_i$, the "best" state to expand is chosen according to: $\max_i \sum_{j=0}^{i-1} \log p(a_j, s_j)$, where $(s_0, a_0), \ldots, (s_{i-1}, a_{i-1})$ is the proof trajectory before state $s_i$ and $\log p(a_j, s_j)$ is the average log probability of the generated tokens for the tactic $a_j$ in state $s_j$.

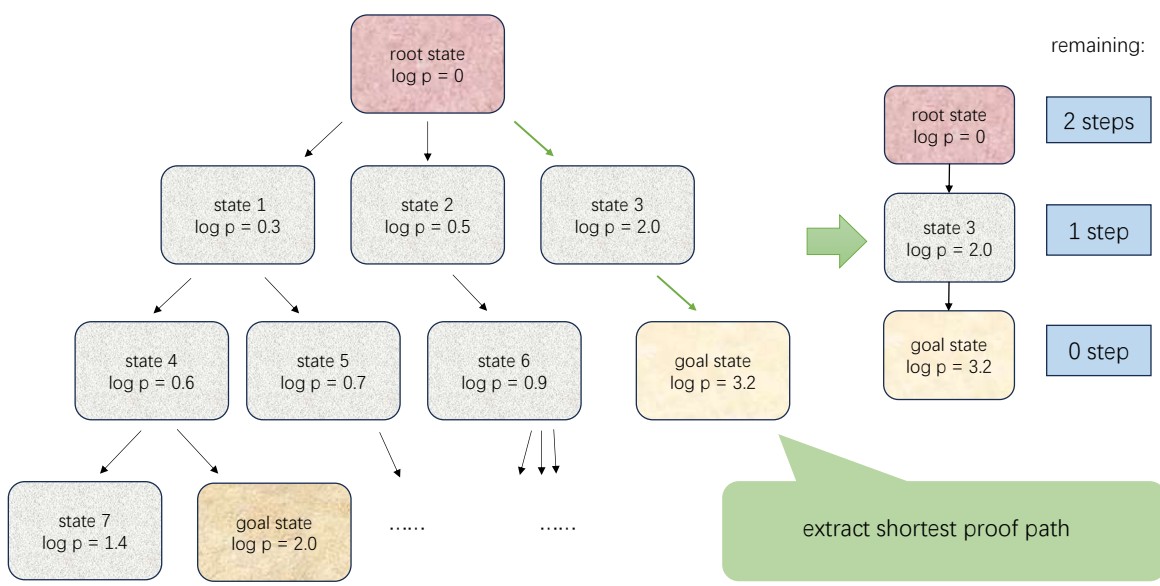

Figure 2: **The visualization of extract proof tree** in theorem proving.

Our work utilizes best-first search in conjunction with the Reprover model (Yang et al., 2024b) to generate successful proof trees. By systematically applying Reprover to all reachable states within a certain depth in a best-first manner, we construct a tree of successful proofs. This approach allows us to collect a dataset of complete proof trajectories, which is then used to train our model to predict the number of remaining steps. This data generation process is crucial for training our model to understand the relationship between proof states and the number of steps required for completion. In particular, if multiple proofs are found for a theorem (i.e., multiple `no_goals` nodes are reached), we select the proof with the minimum depth (the length of the path from the root node to the `no_goals` node) to ensure the quality and consistency of the training data.

Formally, let $T = \{t_1, t_2, ..., t_M\}$ be the set of theorems in our dataset. For each theorem $t_i \in T$, we perform best-first search using the Reprover model to generate a set of successful proof trajectories $P_i = \{p_{i,1}, p_{i,2}, ..., p_{i,k_i}\}$, where $k_i$ is the number of proofs found for theorem $t_i$. Each proof trajectory $p_{i,j}$ is a sequence of proof states: $p_{i,j} = (s_{i,j,1}, s_{i,j,2}, ..., s_{i,j,n_{i,j}})$, where $n_{i,j}$ is the length (number of steps) of the $j$-th proof for theorem $t_i$.

If $k_i > 1$, we select the proof trajectory $p_{i,j^*}$ with the minimum depth: $j^* = \arg\min_j\{n_{i,j} \mid 1 \leq j \leq k_i\}$

Our training dataset $D$ is then constructed by extracting (state, remaining steps) pairs from the selected proof trajectories:

$$D = \{(s_{i,j^*,l}, n_{i,j^*} - l) \mid t_i \in T, 1 \leq l \leq n_{i,j^*}\}$$

where $s_{i,j^*,l}$ is the $l$-th state in the selected proof trajectory $p_{i,j^*}$ for theorem $t_i$, and $n_{i,j^*} - l$ represents the number of remaining steps from state $s_{i,j^*,l}$ to the end of the proof.

### 3.3   Data Balancing

We evaluated our models on a dataset of Lean proofs extracted from Lean Workbook Plus and Mathlib4. The original dataset exhibited a skewed distribution of proof lengths, with an average proof length of $L_{\text{original}} =$

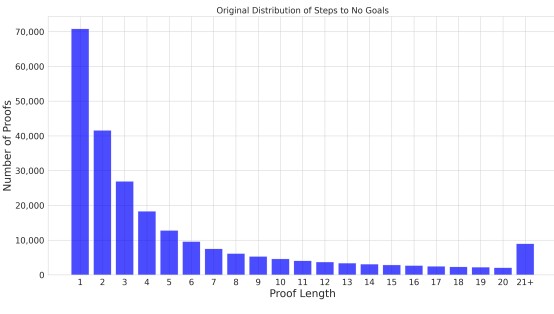
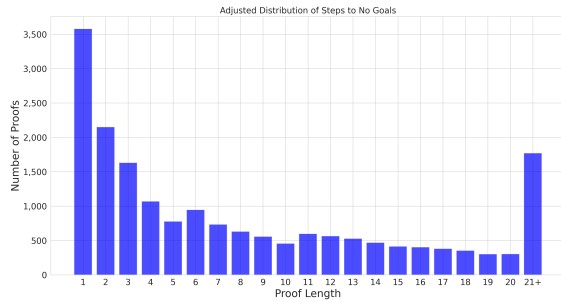

(a) Original dataset of steps distribution

(b) Adjusted dataset of steps distribution by balancing different ranges

Figure 3: Proof-length distributions before and after length balancing. Panel (a) plots *all* collected states prior to balancing (dominated by 1–2-step proofs; average length $L_{\text{original}} = 2.47$). Panel (b) shows the *training set after balancing*, where we down-sample short proofs and (lightly) up-sample longer ones, yielding fewer total states and a higher average length $L_{\text{balanced}} = 10.1$. Bar heights are raw counts, so the totals differ and the $y$-axes are not directly comparable.

2.47. This distribution is shown in Figure 3(a). We adjusted the data distribution based on relative progress within each proof to address this imbalance and ensure a more representative sample of different proof stages. The adjustment was performed by assigning different sampling ratios to five ranges of proof lengths: 1-5 steps (Basic progress, 0.01 ratio), 6-10 steps (Intermediate progress, 0.3 ratio), 11-15 steps (Moderate progress, 0.5 ratio), 16-20 steps (Advanced progress, 0.7 ratio), and 21+ steps (Expert progress, 1.0 ratio). This strategy effectively upsamples longer proofs and downsamples shorter ones, resulting in a more balanced dataset. The resulting adjusted distribution has an average proof length of $L_{\text{adjusted}} = 10.1$, and the comparison is shown in Figure 3(b).

The dataset was then split into training, validation, and test sets. The test set contains 8820 proof states. The training set contains $N_{\text{train}}$ proof states, and the validation set contains $N_{\text{val}}$ proof states. The dataset was partitioned randomly at the theorem level, meaning that all states from a given theorem belong to the same split (either training, validation, or test). This prevents data leakage between splits.

## 4 Model Training & Experiments

This section describes the experimental setup, including model training and results of evaluating Lean-Progress's Step Predictor: Prediction accuracy and search pass rate improvement compared to traditional best-first search via log probability.

### 4.1 Language Model: Remaining Step Predictor

Our approach uses a language model to predict the number of steps remaining to reach a `no_goals` state (proof completion) given a current proof state. While the language model architecture can be varied, we employ a fine-tuned DeepSeek Coder 1.3B model (Guo et al., 2024). This model is trained to predict the number of remaining steps based on the current proof state and, optionally, the history of applied tactics.

The input format for our model is as follows:

`[STATE_BEFORE]state [STEPS_TO_NO_GOALS]steps`

Here, `state` represents the current proof state, encoded as a string representing the current goals. The model is trained to generate the number of remaining steps after this prompt. This input format allows the model to focus specifically on the task of predicting the remaining steps, distinct from predicting the next tactic.

We fine-tuned the DeepSeek Coder 1.3B model on (state, remaining steps) pairs extracted from successful proof trajectories generated using best-first search and the Reprover model as described in the previous

subsection. The DeepSeek Coder model was chosen for its strong performance in code understanding tasks with less than 2B parameters so that personal computers can support inference locally, which we believe translates well to the task of predicting a numerical value representing the remaining steps. The model was fine-tuned using a Mean Squared Error (MSE) loss function with the AdamW optimizer. The training was conducted batch size of 4 and a learning rate of $1e-5$. Other parameters are like betas (0.9, 0.999) with weight decay 0.01 and warmup ratio 0.03. We experimented with other models, such as DeepSeek coder V2 or Prover V1.5, but the model size of 7B is not capable of being used on personal computers. We found that the DeepSeek Coder 1.3B model provided the best balance between performance and computational efficiency.

## 4.2 Proof History Utilization

We investigated the impact of incorporating proof history into the input for remaining step prediction. We compared the performance of our model when using only the current proof state (**state_before**) as input against using both the current state and the preceding tactic sequence (**state_proof**). The results, shown in Table 1, demonstrate the importance of including proof history. The prompt formats used for these two settings are as follows:

- **state_before:** `--- STATE_BEFORE: {state_before}`
  `--- STEPS_TO_NO_GOALS:`

- **state_proof:** `--- STATE_BEFORE: {state_before}`
  `--- PROOF: {proof}`
  `--- STEPS_TO_NO_GOALS:`

Where `{state_before}` represents the current proof state, and `{proof}` represents the sequence of tactics applied so far.

## 4.3 Evaluation

We evaluate our model in two ways. First, we assess the accuracy of our step predictions directly on our generated dataset by calculating the Mean Absolute Error (MAE). Second, we investigate the potential of using our step predictions as a ranking score within a best-first search framework, comparing its performance against standard best-first search based solely on log probabilities.

### 4.3.1 MAE Evaluation on Steps Dataset

To evaluate the accuracy of our step predictions, we calculate the Mean Absolute Error (MAE) on our generated dataset $D$. Given a state $s_{i,j*,l}$ in the selected proof trajectory for theorem $t_i$, our model predicts the number of remaining steps as $\hat{n}_{i,j*,l} = f(s_{i,j*,l})$. The actual number of remaining steps is $n_{i,j*} - l$. The MAE is then calculated as: $\text{MAE} = \frac{1}{|D|} \sum_{(s,n) \in D} |\hat{n} - n|$ where $|D|$ is the total number of (state, remaining steps) pairs in our dataset. This metric provides a direct measure of the average difference between our model's predictions and the true number of remaining steps.

### 4.3.2 Proof History Helps Step Prediction

We evaluate the prediction accuracy using Mean Absolute Error (MAE), which measures the average absolute difference between the predicted number of remaining steps and the actual number of remaining steps. A lower MAE indicates better prediction accuracy. Table 1 presents the MAE and accuracy results for both input formats across different ranges of proof lengths and overall. The table shows the total number of samples for each range, along with the accuracy and MAE achieved by each input format.

From the results in Table 1, we observe a significant performance drop when only the input state is used (**state_before**). Including all previous tactics in the prompt (**state_proof**) provides a "direction" for

| Input | Range | Total Samples | Accuracy (%) | MAE |
|-------|-------|--------------|--------------|-----|
| state | 1–5 | 2,820 | $70.5 \pm 2.1$ | $1.48 \pm 0.18$ |
|       | 6–10 | 1,170 | $58.3 \pm 2.4$ | $3.05 \pm 0.28$ |
|       | 11–15 | 567 | $49.2 \pm 3.1$ | $6.95 \pm 0.55$ |
|       | 16–20 | 563 | $61.8 \pm 2.9$ | $4.25 \pm 0.42$ |
|       | 21+ | 3,700 | $61.1 \pm 2.6$ | $8.45 \pm 0.68$ |
|       | Overall | 8,820 | $62.5 \pm 1.8$ | $5.08 \pm 0.32$ |
| proof | 1–5 | 2,820 | $78.5 \pm 1.8$ | $1.12 \pm 0.15$ |
|       | 6–10 | 1,170 | $62.1 \pm 2.1$ | $2.95 \pm 0.25$ |
|       | 11–15 | 567 | $69.8 \pm 2.5$ | $4.15 \pm 0.35$ |
|       | 16–20 | 563 | $75.4 \pm 2.8$ | $2.85 \pm 0.30$ |
|       | 21+ | 3,700 | $78.2 \pm 2.2$ | $5.05 \pm 0.45$ |
|       | Overall | 8,820 | $75.8 \pm 1.5$ | $3.15 \pm 0.20$ |

Table 1: Comparison of accuracy and MAE (mean $\pm$ std over 3 runs) with and without proof history.

the proof, leading to better performance and more accurate predictions, as evidenced by the consistently lower MAE values across all ranges and overall. This improvement is likely due to the fact that proof history encodes information beyond the current state, such as consistent application of specific mathematical techniques (e.g., repeated use of exponentiation or logarithms) or the overall strategy being employed. This contextual information allows the model to make more informed predictions about the remaining steps.

### 4.3.3 Combining Best-First Search with Steps Prediction

Beyond direct prediction accuracy, we explore the potential of using our step predictions to guide proof search. We integrate our model into a best-first search framework by combining the predicted remaining steps with the log probabilities of the tactic sequence. Specifically, when selecting the next state to expand, instead of using only the cumulative log probability $L(s_i) = \sum_{j=0}^{i-1} \log p(a_j|s_j)$, where $(s_0, a_0), \ldots, (s_{i-1}, a_{i-1})$ is the proof trajectory before state $s_i$ and $\log p(a_j|s_j)$ is the average log probability of the generated tokens for the tactic $a_j$ given state $s_j$, we use a combined score: $C(s_i) = \alpha N(s_i) + (1-\alpha)P(s_i)$, where $\alpha \in [0, 1]$ is a hyperparameter that controls the relative importance of the normalized steps $N(s_i)$ and the log probability $P(s_i)$. The normalized steps are calculated as $N(s_i) = -2\hat{n}_i/N_{\max}$, where $\hat{n}_i = f(s_i)$ is the predicted number of remaining steps for state $s_i$, and $N_{\max}$ is the maximum possible number of steps from all states in a proof.

We compare the performance of this combined approach(where $\alpha = 0.2$) with a standard best-first search using only log probabilities (equivalent to setting $\alpha = 0$). We evaluate both approaches by measuring the number of theorems solved within a fixed number of expansions and the average number of expansions required to find a proof. This comparison demonstrates the effectiveness of incorporating our step predictions into the search process. Proof search requires a search algorithm and a method for interacting with Lean. So, we chose the best-first search for LeanDojo's implementation. Best-first search is parameterized by the maximum number of generated tactics, defined as the number of attempts $\times$ expansion size per iteration $\times$ maximum iterations, subject to a timeout. We use a 2-minute timeout and use beam search with a size of 1 $\times$ 32 due to memory constraints.

We compare our method, which combines predicted remaining steps with log probabilities, against standard best-first search using only log probabilities. We evaluate the LeanDojo v4 test dataset. The primary metric for evaluating proof search performance is the percentage of theorems solved within the timeout. The results of this comparison are shown in Table 2, which demonstrate the effectiveness of incorporating our step predictions into the proof search process.

## 5 Future Work and Conclusion

There are several promising avenues for future work that could further enhance LeanProgress.

| Method | Mathlib4-test Pass Rate (%) |
|---|---|
| Original LogP[†] | $41.4 \pm 0.7$ |
| Steps as Critic[†] | $\mathbf{45.2 \pm 0.6}$ |

Table 2: Proof search pass rates on the Mathlib4-test dataset. We report mean $\pm$ standard deviation over 3 independent runs. Sampling used temperature 0.7; for each problem we sampled 32 candidates once.

**1) Incorporating Tree-of-Thought and Chain-of-Thought Approaches**: One potential direction for future research is to integrate tree-of-thought (ToT) and chain-of-thought (CoT) methodologies into LeanProgress. These approaches could provide a more structured and interpretable way of reasoning about proof progress. By incorporating ToT and CoT, we could potentially improve the model's ability to explain its predictions and provide more detailed insights into the proof process.

**2) Integration with Reinforcement Learning**: A particularly promising avenue for future work is the integration of LeanProgress with reinforcement learning (RL) techniques. LeanProgress's ability to predict the number of remaining steps in a proof can provide a continuous and informative reward signal for RL. Unlike binary rewards only indicating success or failure at the end, this continuous feedback allows the agent to learn from partial progress throughout the proving process. They could also learn efficiently by receiving feedback throughout the proving process while developing better long-term strategies. This could then enable the model to adapt its behavior based on the difficulty and progress of the current theorem and achieve higher success rates.

**3) Lightweight and Scalable Implementations**: Future work could also focus on developing more lightweight implementations of LeanProgress. This could involve exploring model compression techniques or developing more efficient architectures that maintain prediction accuracy while reducing computational requirements. Such improvements would make LeanProgress more accessible and easier to integrate into existing theorem-proving workflows.

In conclusion, we introduce LeanProgress, an approach enhancing interactive theorem proving by integrating a remaining step predictor into the LeanCopilot frontend. Our work makes several contributions to automated theorem proving. We generated a balanced dataset of proof trajectories by adjusting the sampling ratio based on proof length, addressing the challenge of skewed distributions in proof complexity. We then trained a remaining step prediction model using the current proof state and, optionally, the proof history. Integrating this model into the LeanCopilot interface provides users with both tactic suggestions and remaining step predictions, offering a more comprehensive tool for guiding the proof process. Our results highlight the potential of proof progress prediction in enhancing both automated and interactive theorem proving, enabling users to make more informed decisions about proof strategies and bridging the gap between local tactic prediction and global proof trajectory understanding.

# 6    Broader Impact

This work aims to make formal proof development more efficient by providing an interpretable progress signal that can guide search and assist users. Potential benefits include faster iteration in interactive theorem proving, easier onboarding for new contributors, and clearer diagnostics when proofs stall. We have open sourced all code and artifacts with this work. The complete codebase of data generation pipeline, data processing, tool development based on LeanCopilot, and main algorithm are available in the Github repository `https://github.com/lean-dojo/LeanDojo-v2`.

# 7    Acknowledgments

Robert Joseph George is supported by a Caltech Graduate Fellowship. Anima Anandkumar is supported by the Bren Named Chair, Schmidt AI 2050 Senior fellow, and ONR (MURI grant N00014-18-12624).

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

# A  Additional Experiments

## A.1  Sensitivity Analysis: $\alpha$, Temperature, and Sampling Budget

We study the sensitivity of progress-guided best-first search to three key hyperparameters: the mixing weight $\alpha$ (between the step-progress prior and tactic log-probability), sampling temperature, and the number of sampled tactic candidates. Unless otherwise stated, we use $\alpha = 0.2$, temperature $= 0.7$, and 32 samples. We report Mathlib4-test pass rate (mean $\pm$ std over 3 independent runs).

**Effect of $\alpha$ (weight of the progress term).**  We vary $\alpha$ while fixing temperature 0.7 and 32 samples.

| $\alpha$ | Setting | Pass rate (%) |
|---|---|---|
| 0.0 | Pure LogP | $41.4 \pm 0.7$ |
| 0.2 | Default | $45.2 \pm 0.6$ |
| 0.5 | Heavier progress weight | $44.5 \pm 0.5$ |
| 1.0 | Pure Steps | $18.5 \pm 1.1$ |

Table 3: Pass rate vs. $\alpha$. Moderate weighting ($\alpha \in [0.2, 0.5]$) performs best, while relying only on the progress predictor ($\alpha = 1.0$) degrades substantially.

**Effect of sampling temperature.**  We vary temperature while fixing $\alpha = 0.2$ and 32 samples.

| Temperature | Pass rate (%) |
|---|---|
| 0.3 | $43.8 \pm 0.6$ |
| 0.7 | $45.2 \pm 0.6$ |
| 1.0 | $44.6 \pm 0.7$ |

Table 4: Pass rate vs. temperature. Mid-range temperature ($\approx 0.7$) performs best in our setup; lower temperature reduces exploration, while higher temperature adds generation noise.

**Effect of number of samples.**  We vary the sampling budget while fixing $\alpha = 0.2$ and temperature 0.7.

| Samples | Pass rate (%) |
|---|---|
| 8 | $42.9 \pm 0.7$ |
| 32 | $45.2 \pm 0.6$ |
| 128 | $47.2 \pm 0.6$ |

Table 5: Pass rate vs. sampling budget. Increasing samples improves pass rate, with diminishing returns as generations become more correlated and search saturates.

**Model Robustness: DeepSeek Coder 1.3B vs. Qwen 2.5 Coder**

We evaluated our progress estimator with two backbones trained under the same data and optimization settings. Qwen 2.5 Coder yields higher step-prediction accuracy and lower MAE across ranges, which translates into a consistent downstream pass-rate gain.

| | DeepSeek Coder 1.3B | | Qwen 2.5 Coder | |
|---|---|---|---|---|
| **Range** | Acc. (%) | MAE | Acc. (%) | MAE |
| 1–5 | $78.5 \pm 1.8$ | $1.12 \pm 0.15$ | $82.1 \pm 1.6$ | $1.02 \pm 0.12$ |
| 6–10 | $62.1 \pm 2.1$ | $2.95 \pm 0.25$ | $65.4 \pm 2.0$ | $2.75 \pm 0.22$ |
| 11–15 | $69.8 \pm 2.5$ | $4.15 \pm 0.35$ | $72.9 \pm 2.3$ | $3.88 \pm 0.30$ |
| 16–20 | $75.4 \pm 2.8$ | $2.85 \pm 0.30$ | $78.9 \pm 2.6$ | $2.62 \pm 0.28$ |
| 21+ | $78.2 \pm 2.2$ | $5.05 \pm 0.45$ | $81.5 \pm 2.0$ | $4.72 \pm 0.40$ |
| Overall | $\mathbf{75.8 \pm 1.5}$ | $\mathbf{3.15 \pm 0.20}$ | $\mathbf{79.4 \pm 1.4}$ | $\mathbf{2.95 \pm 0.18}$ |

Table 6: Step-prediction with proof history (`state_proof`). Accuracy is exact-match on remaining steps; MAE is mean absolute error in steps. Results are mean $\pm$ std over 3 seeds; overall rows are sample-count weighted across ranges.

| Backbone for Progress Estimator | Mathlib4-test Pass Rate (%) |
|---|---|
| DeepSeek Coder 1.3B (ours) | $45.2 \pm 0.6$ |
| Qwen 2.5 Coder (ours) | $\mathbf{46.4 \pm 0.7}$ |

Table 7: Downstream proof search on Mathlib4-test using identical search settings (temperature 0.7, 32 samples, same $\alpha$ and reranking).

The progress estimator is model agnostic: it takes the same inputs and uses the same loss regardless of backbone, so swapping in a larger or more reasoning-focused model requires no architectural changes to our framework or search layer. We therefore expect strictly better step-prediction accuracy and downstream pass rates as stronger backbones (including specialized reasoning models) are used. In practice, this lets practitioners trade compute for quality without modifying the method itself.

# B  Qualitative Examples

In this section, we showcase LeanProgress successfully predicting the number of remaining steps, in a variety of representative qualitative examples.

Figure 4: Qualitative Example 1/6 of running our Step Predictor on real-world Lean problems.

# C  Practical Tool Development.

With the Step Predictor, one immediate practical application is to couple with tactic suggestion and offer indications of proof progress. While LeanCopilot provides a general framework of developing LLM-based tools natively in Lean, and supports a `suggest_tactic` functionality that offers tactic suggestions, it lacks

Figure 5: Qualitative Example 2/6 of running our Step Predictor on real-world Lean problems.

Figure 6: Qualitative Example 3/6 of running our Step Predictor on real-world Lean problems.

concrete feedback to help users choose among tactic candidates, which creates inefficiency due to repetitive trial-and-error during the theorem proving process. With each tactic candidate, LeanCopilot only offers the resulting state if applying that tactic, together with a log probability score from the tactic generation model. While the log probability score hard to concretize and the resulting state oftentimes too complicated to interpret directly, using prediction of numbers of remaining steps helps guide users directly and concretely in choosing tactics.

Thus, to complement existing tactic suggestion, we leverage LeanCopilot's neural network inference framework in Lean, and builds a practical tool upon `suggest_tactic` that additionally shows the number of remaining steps from each tactic candidate. The whole functionality is wrapped into a single tactic `predict_steps_with_suggestion` that is directly usable within a standard Lean workflow.

This theorem was successfully proven with the aid of our Progress Predictor. A key observation is that a naive application of `norm_num` would not suffice to complete the proof. The Progress Predictor leverages the recorded proof history and inferred the application of the difference of squares factorization. By leveraging proof history and remaining steps, the Progress Predictor likely guided the prover to apply `norm_num` multiple times, ultimately leading to the successful derivation of the target value. A standard Reprover, lacking access to the proof history, would struggle with this theorem.

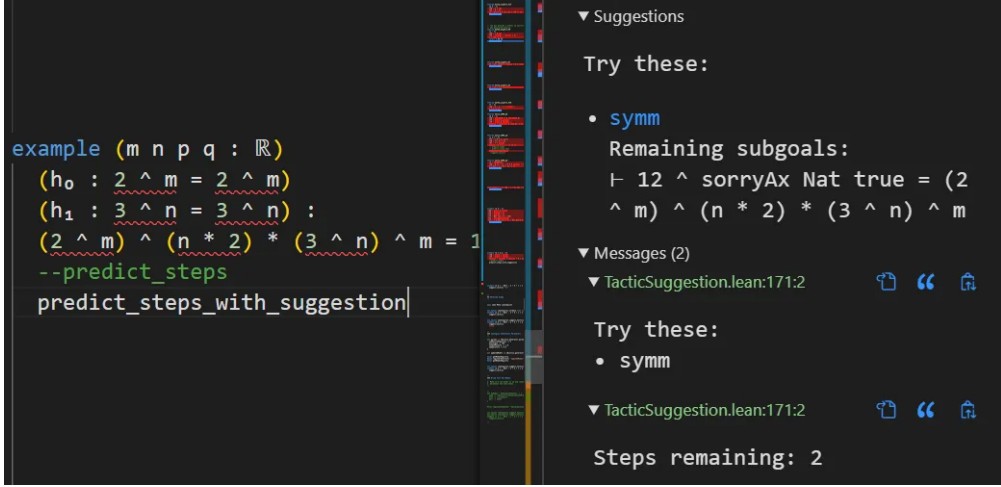

Figure 7: Qualitative Example 4/6 of running our Step Predictor on real-world Lean problems.

Figure 8: Qualitative Example 5/6 of running our Step Predictor on real-world Lean problems.

Figure 9: Qualitative Example 6/6 of running our Step Predictor on real-world Lean problems.

