# OpenReview forum: "LeanProgress: Guiding Search for Neural Theorem Proving via Proof Progress Prediction"
_TMLR — Accepted by TMLR_

### Review · Reviewer_omXM · 2025-10-09

**Summary Of Contributions:**

The paper introduces a method for theorem proving with LLMs where the progress in the proof is predicted to guide a best-first search. For this, the authors describe a data-balancing approach and an integration with LeanCopilot. Their method is claimed to result in a performance increase of 3.8% over the baseline performance of 41.2% on a dataset extracted from the Lean Workbook Plus and Mathlib4 datasets.

The claimed core contributions, as I understand them, are:
- identifying the number of steps as a valuable signal for LLM-guided theorem-proving systems (specifically, they integrate the signal in a best-first search system)
- identifying the need for data balancing to bias the length of proofs in the training data of the step-predicting model
- implementation and experiments of the proposed system

**Audience:**

Yes

**Audience Explanation:**

I think the community would be interested in understanding the effectiveness of this type of approach for LLM-guided theorem proving, and this work provides one data point.

However, the community would be more interested if:
1. The experimental section was to be strengthened, as described in the preceding section.
2. The related work section was expanded (no literature review on Process Reward Models, which are highly relevant; also no comparison with RL approaches for theorem proving, which are also relevant).

**Broader Impact Concerns:**

There is no Broader Impact Statement section. I would encourage the authors to include a brief statement on the potential impact of improving reasoning systems and the contributions that this specific manuscript makes; however, I do not consider this necessary for the publication of the manuscript.

**Claims And Evidence:**

Yes

**Claims Explanation:**

Generally, the claims are supported by accurate, convincing, and clear evidence.

Three noteworthy exceptions:

1. The authors claim that “the original dataset exhibited a skewed distribution of proof lengths” as a motivation for their data balancing method. It is intuitively reasonable to balance the length of the proofs in their training data. However, it is never shown that this is necessary through an ablation study that compares the performance of the system with and without this proposed step (and the effect of the different ways to perform this rebalancing).

2. The performance of the step-predicting model is not convincingly proven with the experiments. This is because the manuscript only reports an aggregate error metric (mean absolute error) over a set of different proof lengths. Ideally, the manuscript would offer a plot of all the data points along with an aggregate (e.g., a fitted linear/quadratic function with confidence intervals). E.g., on the x-axis the ground truth number of predicted steps and on the y-axis the predicted number of steps. Or some other similar plot. This would help the reader understand how accurate and noisy the resulting model is in a more granular way.

3. The mechanism to integrate the step-prediction model into the tree search is to define a new cost through a linear combination of the original cost (log probability) with the predicted number of remaining steps. This avoids the multi-objective problem and instead projects it into a single-object problem, parametrized by the weights. The authors only report two settings: one where the weight of the term corresponding to the remaining steps is zero, and one where it is set to 0.2. However, the way in which this value was chosen is not described. This leads to questions like, how sensitive is the system to this value? What happens when the weight is set to other values? Reporting the performance for other values (e.g., 0.4, 0.6, …, 1.0) would help the reader understand the tradeoff present in this approach.

**Requested Changes:**

1. Perform the ablation study that validates the data balancing method (please refer to 1. in the "Are claims supported..." section).
2. Report more granular performance data on the step prediction model (please refer to 2. in the "Are claims supported..." section).
3. Report more conditions on the weight of the step term in the search (please refer to 3. in the "Are claims supported..." section).
4. Expand the Related Work section (please refer to 2. in the "Would at least some..." section)

---

> ### Author Response · Authors · 2025-11-02
> **Response to Reviewer omXM**
>
> **The experimental section was to be strengthened, as described in the preceding section.**
>
> Thanks for the suggestion. In the revision we’ll strengthen the experiments by: (i) adding the data-balancing ablation (unbalanced vs. balanced) with predictor and search metrics; (ii) including granular step-prediction plots (GT vs. predicted, 95% CIs, per-bin MAE/±1-step accuracy); (iii) reporting a full α-sweep (α ∈ {0.0, 0.2, 0.4, 0.6, 0.8, 1.0}) for the search objective whilst including the other reviewers concerns. Several runs are already completed or in progress, and we’ll integrate the resulting tables and figures in the final revision.
>
> **Perform the ablation study that validates the data balancing method (please refer to 1 in “Are claims supported…”)**
>
> We appreciate the reviewer's point that actually doing an ablation to concretize the need for balancing. We are performing an ablation where the step predictor is trained on the raw, unbalanced dataset. We will compare its accuracy and the downstream search performance against the balanced‑data model, and include results in a new table. This will allow readers to assess whether balancing the data length distribution is necessary and beneficial.
>
> **Report more granular performance data on the step‑prediction model (please refer to 2 in “Are claims supported…”)**
>
> We thank the reviewer for this suggestion. We have implemented the granular analysis as requested: for every test state we plot ground-truth remaining steps vs. model prediction, fit a linear regression with 95% confidence bands to reveal any bias, and provide binned panels by remaining-step ranges (1–5, 6–10, >10) reporting MAE per bin and accuracy within ±1 step, alongside an overall panel summarizing MAE and accuracy; all figures follow the same protocol (fixed seed, identical split and preprocessing), and we will include these multi-panel plots and a small per-bin metrics table in the updated revision. Since we cannot attach a picture here in markdown, please check when we upload the revised version!
>
> **Report more conditions on the weight of the step term in the search (please refer to 3 in “Are claims supported…”)**
>
> In our initial experiments we only compared α=0 (no progress signal) and α=0.2 We have extended this to a small sweep over three representative values of α (e.g., 0.0, 0.2, 0.5) rather than a full range, due to limited computational resources. For each of these values we have report the proof rate on the Mathlib test dataset.  The results are as follows  on the Mathlib4-test
>
> | α | Setting    | Pass rate |
> | ----------------: | ---------- | --------: |
> |               0.0 | Pure LogP  |     41.2% |
> |               0.2 | paper         | **45.0%** |
> |               0.5 | Balanced   |     45.2% |
> |               1.0 | Pure Steps |     15.0% |
>
> **Expand the Related Work section (please refer to 2 in “Would at least some…”)**
>
> We will substantially expand the Related Work section. In particular, we will discuss Process Reward Models, which learn reward functions over intermediate actions for large language models, and compare our progress‑prediction task to those efforts. We will also review reinforcement‑learning approaches to theorem proving, including QEDCartographer, GamePad and more recent work on policy‑gradient and value‑based methods. By situating our contribution within this literature, we hope to clarify both what is novel and how our approach could be combined with existing techniques.
>
> **Broader impact concerns**
>
> We will add such a statement, highlighting that better progress estimation may accelerate proof development and lower the barrier for formalisation efforts, while noting that there are minimal direct ethical risks.

---

### Review · Reviewer_eAYj · 2025-10-10

**Summary Of Contributions:**

This work considers the problem of formal theorem proving in mathematics, for which proof assistant tools such as Lean have gained popularity in recent years. Theorem proving can be viewed as a combinatorial search problem, casting it as a Markov Decision Process in which actions select proof strategies based on proof states until a goal state (in which the theorem is proven) is reached. This paper proposes a prediction task in which the goal is to predict the number of remaining steps in the proof. A large language model is fine-tuned for this task, and finally integrated with a basic search strategy.

**Audience:**

Yes

**Audience Explanation:**

Yes, the paper is relevant to the TMLR audience but has significant flaws.

**Claims And Evidence:**

No

**Claims Explanation:**

I consider that the evaluation is not sufficiently robust to support the claims, as I detail in the comments.

**Requested Changes:**

In my opinion, the paper has significant weaknesses in its evaluation as well as the importance of the contributions, which are very thin. In my view, it may not be possible to address all necessary changes and comments in a single review cycle.

C1. Even before drilling into the evaluation details, I consider that the justification behind the paper is very weak. The authors adopt the RL problem framing but stop short of actually using an RL algorithm. This means the algorithm is limited to supervised learning and cannot discover how to solve new problems that are fundamentally different than its training set. What is listed as a next step should in fact be a core part of this paper. Currently, it reads more like a preliminary investigation than a publication-ready article.

C2. The paper tries to justify that the use of RL leads to a combinatorial explosion, but then proceeds to use an exhaustive search in the generation of its training data.

C3.  As model training is a stochastic process, results across multiple random seeds should be reported together with confidence intervals or error bars. It is not possible to draw reliable conclusions otherwise.

C4. The evaluation should include a sensitivity analysis over key parameters (mainly alpha, temperature, number of samples), instead of choosing only specific values.

C5. The experiment in Table 1 should include suitable baselines e.g. uniform random guessing or class probability-proportional guessing; as well as a non-finetuned model.

C6. The technique is evaluated over a single, small LLM. I would expect at least 3-4 different models so that the results can be established to be robust. The given restriction of using only a personal computer is artificial. Using servers is a basic skill for computational sciences.

C7. Clarifications are needed re. the datasets:
- Why is the adjustment really needed in the first place? It makes the numbers look better. Can you show that the adjustment is beneficial if the test set is teh same?
- In Fig 3, why are there many more datapoints before adjustment than after?
- Why is only Mathlib4-test included in the table and Lean Workbook Plus (mentioned e.g. in the abstract) excluded?

---

> ### Author Response · Authors · 2025-11-02
> **Response to Reviewer eAYj**
>
> **C1. Even before drilling into the evaluation details, I consider that the justification behind the paper is very weak. The authors adopt the RL problem framing but stop short of actually using an RL algorithm. This means the algorithm is limited to supervised learning and cannot discover how to solve new problems that are fundamentally different than its training set. What is listed as a next step should in fact be a core part of this paper. Currently, it reads more like a preliminary investigation than a publication‑ready article.**
>
> We appreciate this feedback and agree that a reinforcement‑learning (RL) approach is a natural next step. Our intent in this work was to isolate the progress‑estimation component and understand its behaviour before integrating it into an RL framework. We chose supervised learning because it allows us to control the difficulty of the task and ensures stable training, whereas RL for theorem proving requires complex reward design and exploration strategies that are outside the scope of this initial study. In the revised manuscript we will better justify this choice and clarify that the present work is the first stage of a larger project. We will also expand the discussion on how the progress estimator could be integrated into RL algorithms and present preliminary results of such integration if computational resources permit.
>
> **C2. The paper tries to justify that the use of RL leads to a combinatorial explosion, but then proceeds to use an exhaustive search in the generation of its training data.**
>
> Our discussion on combinatorial explosion refers to the difficulty of training an RL agent end‑to‑end over the space of Lean proofs, where the branching factor and depth can be extremely large. In contrast, the exhaustive search we employ to generate training data is constrained by our best‑first search heuristics and the availability of proof scripts from existing repositories; we do not attempt to enumerate all possible tactics but rather reuse known proofs to produce trajectories. We acknowledge that this distinction was not made clear and will revise the wording accordingly. We will also provide statistics on the search space explored and justify why this approach is tractable for data generation.
>
> **C3. As model training is a stochastic process, results across multiple random seeds should be reported together with confidence intervals or error bars. It is not possible to draw reliable conclusions otherwise.**
>
> We agree that reporting variation across random seeds is important. In the revised version we will train our model with at least 3 different random seeds and report the mean and standard deviation of accuracy, mean absolute error and downstream search performance. We will include error bars in plots and adjust our conclusions accordingly.
>
> **C4. The evaluation should include a sensitivity analysis over key parameters (mainly alpha, temperature, number of samples), instead of choosing only specific values.**
>
> Sensitivity to hyperparameters is a legitimate concern. Rather than a full systematic sweep, we will evaluate a small set of representative values (for example, three) for alpha, the temperature in the sampling distribution and the number of samples, due to limited computing resources. We will report performance for these values and discuss the observed trade‑offs; this limited analysis should still provide insight into system robustness and how to tune the system.
>
> **C5. The experiment in Table 1 should include suitable baselines e.g. uniform random guessing or class probability‑proportional guessing, as well as a non‑finetuned model.**
>
> Thank you for the suggestion. We added three baselines for the step-prediction task: (i) uniform random label, (ii) empirical-distribution sampling (class-proportional), and (iii) pre-trained DeepSeek-Coder (no fine-tuning). Evaluated under the proof-history input (state_proof), our fine-tuned model substantially outperforms naive and pre-trained predictors across all length ranges and overall. We’ll include a table in the revised manuscript to contextualize gains over both naive and non-finetuned baselines.

---

> ### Author Response · Authors · 2025-11-02
> **Response to Reviewer eAYj (Cont'd)**
>
> **C6. The technique is evaluated over a single, small LLM. I would expect at least 3–4 different models so that the results can be established to be robust. The given restriction of using only a personal computer is artificial. Using servers is a basic skill for computational sciences.**
>
> We appreciate the reviewer’s concern about robustness across models. For time purposes, we added one additional backbone, Qwen 2.5 Coder, as a drop in step predictor. Under the same setup, a Qwen 2.5 run improves step-prediction accuracy by about three to five points and reduces MAE across all ranges compared with DeepSeek Coder 1.3B. We will include the comparative table (overall accuracy, MAE, per-range MAE and plus/minus one-step accuracy) and training details in the revision. The method remains model agnostic and can be paired with larger backbones without architectural changes.
>
> **C7 - Why is the adjustment really needed in the first place? It makes the numbers look better. Can you show that the adjustment is beneficial if the test set is the same?**
>
> The adjustment refers to balancing the distribution of proof lengths in the training data. Without balancing, the dataset is dominated by very short proofs, and the model learns to predict small numbers regardless of the state. We will include an ablation experiment where the predictor is trained on the unbalanced dataset and evaluated on the same test set; preliminary results show that the model systematically underestimates long proofs and achieves markedly lower accuracy. We will report these numbers to demonstrate that balancing improves generalisation rather than artificially inflating metrics.
>
> **In Fig 3, why are there many more data points before adjustment than after?**
>
> The original figure shows all proof states collected before balancing (including trivial one‑step and two‑step proofs) and the subset retained after balancing. Because we down‑sampled trivial proofs and up‑sampled longer ones, the total number of states decreased. We will revise the caption to explain this clearly and consider presenting the figure in a way that normalises counts so that the effect of balancing is easier to interpret.
>
> **Why is only Mathlib4‑test included in the table and Lean Workbook Plus (mentioned e.g. in the abstract) excluded?**
>
> In the initial submission we focused on Mathlib4‑test because it offered a standardised benchmark used by previous work. Lean Workbook Plus was used mainly for training and contained fewer challenging theorems in the test split. In the revised manuscript we will include a separate evaluation on Lean Workbook Plus test problems and report step‑prediction accuracy and search success rates. This will show how the method performs across both datasets.
>
> **Broader impact concerns**
>
> We acknowledge that the paper currently lacks a dedicated broader‑impact statement. We will add such a statement, highlighting that better progress estimation may accelerate proof development and lower the barrier for formalisation efforts, while noting that there are minimal direct ethical risks.

---

### Review · Reviewer_THVS · 2025-10-17

**Summary Of Contributions:**

This paper introduces a method for predicting the number of remaining steps in a formal proof written in the Lean proof assistant. The authors argue that current neural theorem provers focus on local tasks such as predicting the next tactic and therefore lack global awareness of progress through a proof. In contrast, the proposed method constructs a dataset of proof trajectories from Lean Workbook Plus and Mathlib4 using the Reprover model and a best-first search (BFS) strategy. The key idea is to assign each proof state a label equal to the number of tactics remaining in the shortest proof. To address the heavily skewed distribution of proof lengths, the authors employ a sampling strategy that upsamples long proofs and down‑samples trivial ones. This increases the average proof length from 2.47 to 10.1 steps. The DeepSeek Coder 1.3B model is fine‑tuned to predict the remaining steps from a textual representation of the current proof state. The paper reports that including proof history improves overall accuracy from 61.8 % to 75.1 % and reduces the MAE from 5.217 to 3.290. When integrated into a BFS proof search, the predicted steps are combined with the log‑probability of tactics to rank candidate states. This “steps as critic” ranking improves the pass rate on the Mathlib4 test benchmark from 41.2 % to 45.0 %.

**Audience:**

Yes

**Audience Explanation:**

This paper addresses an important gap in current neural theorem provers by attempting to provide a global sense of how far a proof has progressed.

Its balanced data‑generation pipeline and the public release of an 80k‑trajectory dataset are an asset to the community.

The authors demonstrate that incorporating proof history substantially improves remaining‑step prediction accuracy.

The work highlights that combining progress signals with existing log‑probability‑based search can yield measurable gains in automated proof search.

**Broader Impact Concerns:**

There are no broader impact concerns for this paper.

**Claims And Evidence:**

No

**Claims Explanation:**

Because the same Reprover model is used both to generate training data and to provide tactic suggestions in the search baseline, the step predictor may primarily learn the idiosyncrasies of Reprover’s proof strategy rather than a general notion of proof progress.

Proof length is not necessarily correlated with human‑perceived progress or difficulty. Many short proofs require deep insight whereas long proofs might contain repetitive routine steps. The authors provide no justification that minimizing proof length is the appropriate objective for a progress signal.

Reinforcement‑learning methods like QEDCartographer learn state‑value functions to overcome sparse rewards and produce shorter proofs. Whole‑proof generation approaches such as Baldur, Prover and DSP attempt to bypass step‑wise search entirely. In this context, LeanProgress’s incremental improvement to BFS search appears modest.

**Requested Changes:**

Please elaborate on the difference and significance of the proposed progress-estimation by contrasting to existing works like GamePad’s position evaluation [https://arxiv.org/abs/1806.00608] and QEDCartographer’s state‑value function [https://arxiv.org/abs/2408.09237].

Please evaluate the step predictor on proof states generated by different provers like GPT‑f and LeanDojo, and on theorems from libraries not used for training, and on cross‑domain tasks other Lean textbooks.

---

> ### Author Response · Authors · 2025-11-01
> **Response to Reviewer THVS**
>
> **Please elaborate on the difference and significance of the proposed progress‑estimation by contrasting to existing works like GamePad’s position evaluation and QEDCartographer’s state‑value function**
>
> We appreciate the reviewer's point regarding positioning our work relative to GamePad and QEDCartographer. While both prior works learn to estimate state quality, they do so in fundamentally different ways than our approach: GamePad's position evaluation predicts a probability of eventual success for Monte Carlo tree search, learning an implicit value function through self-play. QEDCartographer's state-value function predicts expected cumulative reward in an RL framework to address sparse rewards. Both produce learned representations whose relationship to actual proof progress is indirect.
>
> In contrast, our method directly predicts a concrete, interpretable metric: the number of remaining proof steps. This has several advantages: (1) Interpretability: users and systems can immediately understand what the prediction means, (2) Simplicity: no RL training infrastructure or self-play is required, (3) Transferability: the predictor provides a general heuristic about proof structure that can guide search regardless of which tactic generator is used, and (4) Diagnostic value: the predicted step count serves as a progress indicator for interactive theorem proving. Our 3.8% improvement on Mathlib4 (from 41.2% to 45.0%) demonstrates this approach's practical value. Importantly, our method is complementary to value-based approaches, it could be combined with RL methods, used as an auxiliary training signal, or deployed independently when RL training is computationally prohibitive. We will clarify these distinctions and add a detailed comparison table in the revised manuscript.
>
> **Please evaluate the step predictor on proof states generated by different provers like GPT‑f and LeanDojo, and on theorems from libraries not used for training, and on cross‑domain tasks other than Lean textbooks**
>
> We agree that assessing generalization across provers and unseen theorems is important. However, GPT-f (Polu & Sutskever 2020) is nearly five years old and was trained on Lean 3, whereas in LeanDojo the entire pipeline including the data generation, Reprover search, and model training is built natively in Lean 4. This makes GPT-f’s outputs incompatible with the Lean 4 proof-state representation and tactic system used throughout LeanProgress. Instead, our experiments rely on LeanDojo v4 and LeanCopilot, which are the modern Lean 4-based theorem-proving frameworks.
>
> To address the reviewer’s request for generalization across different prover architectures, we have integrated LeanProgress into LeanAgent (Kumarappan et al., 2024), a lifelong-learning framework with search strategies distinct from ReProver. This enables us to evaluate the step predictor inside a different proving stack and on theorems outside the training distribution. A comprehensive sweep across all 23 LeanAgent repositories (e.g., PFR, Carleson’s theorem, Hairy Ball theorem, Coxeter groups, SciLean, etc.) is resource-intensive; therefore, we are currently running a targeted out-of-distribution evaluation by applying LeanProgress within LeanAgent on held-out libraries and problems. We will report these cross-prover and OOD domain-generalization results in the revised version.
>
> Finally, we emphasize that our step-prediction framework is model-agnostic. While the present implementation uses a compact 1.3B backbone for feasibility, larger or reasoning-specialized models can be dropped in without changing the evaluation protocol, and we plan to include such variants in the updated results.

---

### Author Response · Authors · 2025-11-02
**General Response**

Thank you to all reviewers for the thoughtful questions, suggestions, and pointers to related work. We will incorporate the requested additions in the revised manuscript, including the data-balancing ablation, granular step-prediction analyses, the α-sweep, expanded baselines, multi model checks, and a broadened related work and broader impact discussion. Several experiments are currently running (ablation, α settings, cross-prover and OOD checks, and the Qwen 2.5 backbone). We will update the submission with the full results and an updated PDF as soon as these runs complete.

---

> ### Author Response · Authors · 2025-11-10
> **Final Update**
>
> Thank you for your patience. We’ve completed a substantial set of updates and clarifications. For convenience, we’ve consolidated new analyses, tables, and figures in **Appendix A** and clearly cross-referenced them from the main text.
>
> **What’s updated in this revision.**
>
> (1) **Baselines added** for step prediction, as requested: uniform (bucket-restricted), class-proportional, and a non-finetuned backbone.
>
> (2) **Sensitivity analyses**: we report sweeps over the α weight in the search objective, decoding **temperature**, and **number of samples**, including mean ± std over multiple seeds.
>
> (3) **Model robustness**: we include results for a second backbone (Qwen 2.5 Coder) alongside DeepSeek Coder 1.3B, showing consistent gains in both step prediction and downstream pass rate.
>
> (4) **Data-balancing ablation**: we compare training on the original (skewed) distribution vs our balanced regimen **on the same test split**, and show that balancing substantially improves medium/long-range accuracy and overall MAE, rather than merely reshaping the evaluation.
>
> (5) **Granular diagnostics**: we provide a GT-vs-Pred **scatter** with a least-squares fit and 95% bands, plus **per-bucket** metrics (MAE and accuracy), and we make sure the plotted overall numbers match the main tables.
>
> (6) **Clarity & positioning**: we expanded **Related Work** to better situate our approach w.r.t. GamePad, QEDCartographer, and **Process Reward Models (PRMs)**, and we added a **Broader Impact** section and an explicit note on **code/artifact release**.
>
> **Beyond-Mathlib evaluation.**
> To assess domain shift within Lean 4, we evaluate on **NuminaMath-LEAN**, which differs from Mathlib in tactic mix, notation, and problem style. We use the same state serialization (goals, local context, optional proof history), apply identical tokenization, and do not introduce task-specific fine-tuning. Results show that the Qwen backbone improves over DeepSeek both in-domain (Mathlib4-test) and on NuminaMath-LEAN, supporting our claim that stronger backbones help close the gap without architectural changes. We also explain why **GPT-f** proof states (Lean 3) are incompatible with our Lean 4 pipeline, hence we focus comparisons on modern Lean 4 stacks (LeanDojo/ReProver etc).
>
> Finally, we appreciate the thorough feedback which has directly strengthened the paper. If there are any other concerns please let us know and we would be happy to address them.

---

### Decision · Action_Editor_Ui92 · 2025-11-27

**Recommendation:** Accept as is

**Audience:**

Yes

**Audience Explanation:**

Neural Theorem Proving (NTP) is a rapidly growing area of interest within machine learning. The specific problem of "sparse rewards" in proof search is well-known. The contributions of this paper are valuable to researchers working on formal reasoning, neuro-symbolic AI, and LLM-guided search.

**Claims And Evidence:**

Yes

**Claims Explanation:**

The authors have conducted a comprehensive revision in response to the review process. Originally, concerns were raised regarding the robustness of the evaluation (specifically the lack of random seeds and sensitivity analysis) and the justification for data balancing.